# Disclosing Strain: How Psychosocial Risk Factors Influence Work-Related Musculoskeletal Disorders in Healthcare Workers Preceding and during the COVID-19 Pandemic [note 1]

**DOI:** 10.3390/ijerph21050564

**Published:** 2024-04-29

**Authors:** Carla Barros, Pilar Baylina

**Affiliations:** 1Faculty of Human and Social Sciences, University Fernando Pessoa, RISE-UFP, FP-I3ID, 4249-004 Porto, Portugal; 2School of Health, Polytechnic of Porto, 4200-072 Porto, Portugal; pilarbaylina@ess.ipp.pt; 3School of Medicine and Biomedical Sciences, University Fernando Pessoa, 4249-004 Porto, Portugal; 4RISE-UFP, University Fernando Pessoa, 4249-004 Porto, Portugal; 5I3S—Instituto de Investigação e Inovação em Saúde, 4200-135 Porto, Portugal

**Keywords:** psychosocial risk factors, work-related musculoskeletal disorders, healthcare workers, nurses, age, pre-pandemic, pandemic

## Abstract

Healthcare workers, particularly nurses, engage in a daily work routine that takes a toll on their emotional well-being, rendering them vulnerable to psychosocial risk factors. This research seeks to analyse the influence of psychosocial risk factors on the occurrence of work-related musculoskeletal disorders among nurses. An additional analysis was performed to understand the role of age in work-related musculoskeletal disorders and the perception of psychosocial risk factors. The study was conducted during two separate periods—pre-pandemic and pandemic times—involving a sample of 456 nurses from both public and private hospitals in Portugal. The INSAT—Health and Work Survey—was used as measuring instrument. The primary observations indicated a consistency between psychosocial risk factors and the occurrence of work-related musculoskeletal disorders. The findings revealed a significant exposure to psychosocial risk factors, with work pace, intensity, work relationships, and emotional demands exhibiting higher global average percentages during both periods, pre-pandemic and pandemic. Nonetheless, we find that the psychosocial risk factors change when we analyse the pre-pandemic and pandemic results. During the period before the pandemic, the psychosocial risk factors that were most commonly reported included the demanding pace of work, long working hours, and emotional demands. Through the pandemic, the most pronounced psychosocial risk factors were work relationships, employment relationships, and ethical and values conflicts. Therefore, research in this domain is essential to understanding psychosocial risk factors and assessing the less obvious links between work and health.

## 1. Introduction

The COVID-19 pandemic caused new challenges for work activity, with healthcare emerging as a prime example of one of the most impacted sectors [1]. Significant changes in the workplace context due to a stricter safety protocol led to several adjustments in how healthcare services are delivered and how healthcare workers work within their work environments. This situation led to different behavioural responses and an increase in physical and psychological demands, and working long hours under intense pressure to provide care to those affected with a high emotional strain changed the usual way of working [2,3,4]. In fact, crises and emergency situations challenge the health and psychological wellbeing of individuals, and healthcare workers are no exception. Several responsibilities and situations that make them deal with pain, suffering, deaths, and losses under inadequate work circumstances can generate high levels of stress, emotional effort, and physical exhaustion. This is particularly true as demonstrated by the difficulties brought up by the most recent COVID-19 pandemic [5,6,7].

Owing to the unique nature of their job, as well as the features of their workplace and organizational structure, healthcare workers—and nurses in particular—are more likely to be at risk for both physical and mental health issues [8,9,10]. In fact, nurses are among the most vulnerable professional groups; they frequently experience medical and psychological disorders that are made worse by the demanding nature of their jobs, which includes long shifts, intense work, and emotional stress [11,12]. These professionals’ well-being has been badly influenced by difficult situations they have been put in due to a lack of support systems, patient care, high demand, constant time availability, longer working hours, and shift work. These risk variables that adversely interfere with work activities and jeopardize workers’ capacity to conduct job tasks embrace psychosocial risk factors.

Psychosocial risk factors affecting health and well-being include workload and time pressure, role ambiguity and conflict, work environment, organizational culture and support, harassment and work violence, employment insecurity, and social support and connections [13,14,15]. Furthermore, research revealed that during the pandemic, exposure to psychosocial risk factors increased. These risks included high emotional demands, time constraints, organizational challenges at work, and increased workloads. Additionally, there was a lack of support from management and staff, which had a serious negative impact on health, specifically work-related musculoskeletal disorders [16,17,18].

Studies on the relationship between psychosocial risk factors and work-related musculoskeletal disorders have revealed that factors such a high workload or a lack of social support can contribute the onset of these disorders [18]. Certain occupational categories have a higher prevalence of work-related musculoskeletal disorders than others. The demands and physically and emotionally taxing nature of their work make healthcare workers more susceptible to this health problem. Musculoskeletal disorders have been linked to high job demands, including long working hours, strict scheduling, emotional pressures, and workplace conflicts [19,20,21].

Studies have been developed showing that nursing is one of the professions most affected by work-related musculoskeletal disorders [22,23,24]. During the COVID-19 pandemic, the increase in the intense pace of work, the contact with suffering and death, the emotional overload of work, and the associated time pressure and shift work are psychosocial risk factors related to higher risk of developing musculoskeletal disorders [25,26].

In fact, there is a relationship between psychosocial variables and work-related musculoskeletal disorders in nursing, namely high psychological stress, inadequate social support, and effort–reward imbalance [27,28]. Nurses are exposed to an excessive number of patients assigned to each professional, increasing the physical and mental overload, forcing them to work at an intense pace, aggravated by unclear instructions and insufficient time to complete the task [29]. In addition to these factors, the lack of means and resources to carry out a *good* job increased with COVID demands and triggered more physical and mental health effort and, more specifically, the expansion of musculoskeletal disorders [11].

The age dimension plays an important role in the comprehension of work-related musculoskeletal disorders [18]. For this reason, it is necessary to use specific indicators that allow a more contextualised analysis; in fact, the age variable can also better explain the characteristics of work that most influence musculoskeletal disorders. The ageing process, viewed as a multifaceted and intricate phenomenon encompassing biological, psychological, and social dimensions, is inherently connected to exploring the relations and the effects of work-related factors. Considering that the ageing process can advance due to the working conditions in which work activity is developed, it is important to analyse the role of age in the development of musculoskeletal disorders. Results showed associations between the prevalence of work-related musculoskeletal disorder symptoms and individual characteristics, such as age; the majority of the studies revealed that the prevalence of musculoskeletal complaints is known to be a predisposing factor at old age compared to younger nurses [30,31,32].

Hence, musculoskeletal disorders may arise as a result of exposure to psychosocial factors. Has the pandemic made these health issues worse? Have the relations between psychosocial risk factors and musculoskeletal disorders changed or evolved? What is the role of age? Thus, the first objective of this study is to analyse the associations between the psychosocial risk factors and work-related musculoskeletal disorders and the second objective is to analyse the associations between age and work-related musculoskeletal disorders and psychosocial risk factors, both between pre-pandemic and pandemic times.

## 2. Materials and Methods

### 2.1. Study Design and Ethics

A cross-sectional study was performed with Portuguese nurses working in both public (3) and private (2) hospitals participating, and it was carried out at two different times: before and during COVID-19. All participants gave their informed agreement to contribute, participation was completely optional, and confidentiality and anonymity were ensured. The Ethics Committee of Fernando Pessoa University granted approval for the study protocol (Ref. PI-112/19), adhering to all procedures outlined in the Declaration of Helsinki.

### 2.2. Data Collection and Measures

#### 2.2.1. Data Collection

Data were collected using INSAT—Health and Work Survey, a self-administered questionnaire. Instructions for filling out the questionnaire, a cover paper and pencil, and a questionnaire were followed by the researchers’ guidance outlining the survey’s goal. The sample included 456 nurses in Portugal: 172 nurses from September to November of 2019 prior to the pandemic and 284 nurses from May to July of 2021 during pandemic times. Each nurse received an envelope from the Human Resources department containing details about the study’s objectives and the tools employed in the study protocol. Subsequently, the completed materials were returned in a sealed envelope.

#### 2.2.2. Measures

INSAT—Health and Work Survey, an instrument that assesses risk factors, health issues, and working conditions, provided support for this investigation [33]. In this study, the psychosocial risk scale, the work-related musculoskeletal disorders variable, and the age variable were used to answer the objectives of the current investigation. The psychosocial risk scale has different risk categories, work pace and intensity and working hours, which encompass the notion of psychological demand and effort. The amount of work corresponds, on the one hand, to the time it takes up (by its duration and/or the organisation of working time) and, on the other hand, to the intensity of the work and its complexity. Emotional demands are related to the need to manage your own emotions and the emotional state of the people you interact with in the world of work. Emotional demands are felt above all in relationships with the public, be they clients, users, patients, or others. The quality of relationships with managers and co-workers at work and employment relations with the organisation encompass relations between workers as well as relations between the worker and the organisation that employs him or her. These relationships should be analysed in the light of the concepts of integration, recognition, and sense of justice, and ethical and values conflicts are experienced when, in the work situation, the worker is forced to act against their professional, social, or personal values. The items in these categories are arranged differently. The psychosocial risk factors are evaluated with an ordinal scale with a 6-point Likert scale from 0: not exposed to 5: being exposed with high discomfort. The aim of this study was to assess whether psychosocial risk factors were related to musculoskeletal disorders and not the degree of discomfort they caused. Therefore, we decided to transform the ordinal scale into a nominal one, classifying only the fact that the risk factor was not present (0) or was present (1).

The work-related musculoskeletal disorders variable used was measured by “musculoskeletal disorders not related to my work activity”, “musculoskeletal disorders caused by my work activity”, and “musculoskeletal disorders aggravated by my work activity”. This variable was transformed to a nominal variable with binary scale, 0 for “not related to my work activity” and 1 for “caused by my work activity” and “aggravated by my work activity”. Regarding psychometric qualities, INSAT exhibits strong internal consistency, as demonstrated by a Rasch Partial Credit Model (PCM) analysis, with a reliability coefficient > 0.8. According to the partial credit model, every item has a unique structure for its rating scale. It comes from multiple-choice exams where answers that are not quite right but show some understanding are partially counted toward a right answer. Each item has a different level of partial accuracy [33].

In this sense, risk integrates the relationship between exposure to risk factors on one hand and effect on the other. From this perspective, the idea of measuring the effect from psychosocial risk factors should be organised since they are not specific and measurable effects. It is important to highlight those disorders/problems of mental, physical, and social health, which most of the time do not have immediate effects; they are cumulative and deferred over time, namely as work-related musculoskeletal disorders (WMSDs).

### 2.3. Statistical Analysis

We performed descriptive statistics to provide an overview of the sample across all evaluated variables. Psychosocial risk factors were subjected to frequency and percentage analysis. To analyse the impact of exposure to psychosocial risk factors on musculoskeletal disorders, a logistic regression was performed. The psychosocial risk factors (ordinal variable with a Likert scale from 0: not exposed to 5: being exposed with high discomfort) and work-related musculoskeletal disorders (nominal variable, no: 0; yes: 1).

Prior to the regression analysis, psychosocial risk factors were transformed into a nominal scale (no: 0; yes: 1). This transformation was carried out as the study aimed to identify the effect of exposure rather than assessing the degree of discomfort associated with that exposure. After this, a logistic regression analysis was performed to identify associations between musculoskeletal disorders and psychosocial risk factors in pre-pandemic and pandemic times. The adherence to the assumptions of the method was confirmed, ensuring the reliability of the obtained results.

Then, a point biserial correlation analysis was performed to better understand the relationship between age and work-related musculoskeletal disorders and the relationship between age and psychosocial risk factors. Again, the adherence to the assumptions of the method was confirmed, ensuring the reliability of the obtained results.

The data analysis was conducted using the IBM SPSS statistical program for Windows, specifically version 28.0 (SPSS Inc., Chicago, IL, USA). All computations were carried out with a 95% confidence interval, and statistical significance was set at *p* < 0.05.

## 3. Results

### 3.1. Participants

A total sample of 456 nurses working in public and private hospitals in Portugal’s north and central regions participated in this study, 172 nurses in pre-pandemic times (September to November 2019) and 284 in pandemic times (May to July 2021) (Table 1).

Regarding the contract type, in the pre-pandemic period, 91% of the participants worked under permanent or open-ended contracts, 95% had full-time working hours, and 75% worked on rotating shifts. In pandemic times, the sample was very similar, with almost 91% of the participants working under permanent or open-ended contracts, more than 93% having full-time working hours, and more than 65% working on rotating shifts.

### 3.2. Descriptive Analysis

The descriptive analysis of the INSAT survey (Table 2) illustrates the distribution of “yes” responses concerning psychosocial risk factors in the workplace that substantially affect the professional practice of nurses. The findings indicate a notable prevalence of psychosocial risk factors, with work pace, intensity, and emotional demands emerging as prominent risk factors, characterised by higher overall mean percentages.

Table 3 presents the descriptive analysis of the work-related musculoskeletal disorders. As can be seen, more than 50% of participants had this sort of disorder in both circumstances, and over 60% of the participants (62.3% in pre-pandemic and 80% in pandemic times) agreed that work activity was the cause or aggravation of these problems.

The results showed that “musculoskeletal disorders aggravated by my work activity” increased, while “musculoskeletal disorders not related to my work activity” decreased in pandemic times.

### 3.3. Inferential Analysis

A logistic regression analysis was performed to investigate the relationship between psychosocial risk variables and work-related musculoskeletal disorders (WMSDs). For this, the variable was transformed to a binary scale, 0 for “not related to my work activity” and 1 for “caused by my work activity” and “aggravated by my work activity”. Compliance with the method’s assumptions was confirmed in the conducted study, and the outcomes were deemed reliable. Table 4 presents the findings (only for *p* < 0.05).

Three important psychosocial risk factors have considerably raised the perception of musculoskeletal disorders before the pandemic, as Table 4 illustrates. The factor “managing contradictory instructions” enlarged the perception of musculoskeletal disorders more than fifteen times (16.940; C.I. 95%: 3.289–87.249), the factor “simulating good mood and/or empathy” increased more than five times (5.480; C.I. 95%: 1.526–19.684), and the factor “exceeding normal working hours” increased almost four times (3.892; 95% CI 1.745–18.324).

In pandemic times, as shown in Table 4, four key psychosocial risk factors significantly increased the perception of musculoskeletal disorders. The risk factor “being afraid of suffering a work-related injury” increased almost five times (4.611; C.I. 95%: 2.090–10.172). “my organization disregards my well-being” increased the perception of musculoskeletal disorders more than four times (4.058; C.I. 95%: 1.703–9.673), “needing help from colleagues and not having it” increased the perception of musculoskeletal disorders almost four times (2.994; C.I. 95%: 1.453–6.170), and finally, the risk factor “lack of necessary means to perform a good job” increased the perception almost three times (2.994; C.I. 95%: 1.453–6.170).

A more careful analysis of the psychosocial risk factors’ dimensions shows a change in paradigm. Before the pandemic times, all statistically significant psychosocial risk factors were from dimensions such as work pace and intensity, working hours, and emotional demands. But, during the pandemic times, this change and all statistically significant psychosocial risk factors were from dimensions such as employment relationships, ethical and values conflicts, and working relationships.

After this analysis, a correlation between age and work-related musculoskeletal dis-orders was performed using the point biserial correlation coefficient (Table 5).

The coefficients presented are low (and negative), but only in the case of the pandemic times is statistical significance verified. This means that with the increased of age workers, there is a decrease in the perception of work-related musculoskeletal disorders.

To complement this analysis, a point biserial coefficient analysis was performed between psychosocial risk factors and age and values and is presented in Table 6 (only for *p* < 0.05).

Again, the correlation coefficients found are low and negative; this means that with the increase in the age workers, there is a decrease in psychosocial risk factor perception. A more careful analysis shows that in pre-pandemic times, only psychosocial risk factors from the work pace and intensity dimension were statistically significant. When analysed, the pandemic times psychosocial risk factors from other dimensions, such as employment relationships and emotional demands, appeared as statistically significant. The psychosocial risk factor “hyper-solicitation” is the only one that appeared in pre-pandemic and pandemic times.

## 4. Discussion

Nursing is a physically and emotionally exhausting work activity that became more demanding with the pandemic, increasing vulnerability to physical and mental health problems [9,34]. In fact, among all other professions, nursing has the highest potential for developing work-related musculoskeletal disorders (WMSDs). Frequently, nurses work long shifts, often in high-stress environments, which can further exacerbate the risk of developing WMSDs. Studies have revealed that nurses’ experiences with musculoskeletal pain are not only related to the physical factors associated with the exercise of their activity but can be related to psychosocial factors: high levels of stress may lead to increased muscle tension, fatigue, and overall discomfort, and also, inadequate support systems within healthcare settings can further compound the effects of physical strain experienced by nurses [8,28,29].

Therefore, research shows a significant presence of psychosocial risk factors: work intensity, emotional demands related to stress with patients, having to deal with serious situations, and the lack of organizational support leveraged by the COVID-19 pandemic have potentiated the worsening of work-related musculoskeletal disorders [9,10,35]. Studies [17,19,27] reveal a consistency relation between psychosocial risk factors and work-related musculoskeletal disorders. A set of psychosocial risk factors, mainly work pace and intensity, social relationships, and emotional demands, may forecast the presence of musculoskeletal disorders. In fact, increased work pace and intensity, lack of adequacy in the organization of working hours accompanied by the lack of support and resources increased nurses’ health problems associated with musculoskeletal pain.

However, distinctions emerged in the array of psychosocial risk factors that predominantly impacted the onset of work-related musculoskeletal disorders in the pre-pandemic vs. pandemic periods. Prior to the pandemic, there was a greater emphasis on emotional demands, long workdays, and an intense work rhythm. During the pandemic, working relations, employment relations, and ethical and values conflicts emerged as the most influential factors. In addition to the typical psychosocial risk factors linked to nurses’ work, such as emotional demands and the pace and intensity of their tasks, additional forms of risks surfaced during the pandemic. Factors like insufficient support and resources and fear of contagion, coupled with inadequate means to carry out quality work, escalated during the pandemic, contributing to heightened psychological stress among nurses and an increased incidence of work-related musculoskeletal disorders [17,21,25,26].

Indeed, when contrasting the findings between the pre-pandemic and pandemic periods, noticeable shifts in psychosocial risk factors become apparent. In the pre-pandemic period, the frequently reported psychosocial risk factors were the intense pace of work, working hours, and emotional demands. Conversely, during the pandemic, the predominant psychosocial risk factors reported were related to work relationships, employment relationships, and ethical and values conflicts. The results found for age also revealed differences between the pre-pandemic and pandemic times. When analysed, the relation between age and work-related musculoskeletal disorders was only found to be statistically significant in pandemic times. When analysing the relation between age and psychosocial risk factors, a change in paradigm was also found emerging in new dimensions that affect nurses during work activity in pandemic times. But even so, we cannot fail to mention studies that have already demonstrated the emergence, with age, of strategies that, acquired with experience and accumulated knowledge, help the “older” worker to cope with the constraints imposed by the job, which are therefore no longer perceived as difficulties [36,37,38]. It appears that nurses had grown accustomed to the initial set of risks, considering them intrinsic to their work while simultaneously allowing the emergence of a different category of psychosocial risk factors. Developing research targeted at controlling and preventing work-related musculoskeletal disorders in nurses requires an understanding of these psychological variables, efforts to improve social support at work, and measures to reduce the effort–reward imbalances through equitable systems of recognition and remuneration are a few examples of strategies. Certainly, gaining a more profound comprehension of the psychosocial risk factors enables more effective intervention strategies, encompassing both the individual and organisational levels in order to prevent issues and foster health and well-being within the workplace environment.

## 5. Limitations and Future Research

A cross-sectional study with a larger sample size and a longitudinal design in future research can render it possible to determine more robust causal relationships between the variables. Given that it is now impossible to carry out a longitudinal study related to the COVID-19 pandemic, it is important to emphasise that a longitudinal design would be a more appropriate approach in the event of another type of pandemic.

The constraints of the sample not being the same before and during the pandemic could be explained by the difficulties experienced in approaching health professionals during these periods (always overloaded) as well as the high turnover of these posts, which led to many replacements during the periods under analysis. Therefore, in future work, this type of situation will be avoided to ensure a better and more appropriate analysis of the results. This study used self-reported questionnaires, which have known limitations.

Therefore, future research would benefit from combining qualitative methods to gain more depth and confirmation. The sampling method can include healthcare professionals from different categories (physicians, nurses, physiotherapists, dentists, clinical psychologists, ambulance workers, health care assistants, and health supporters, among others), as embracing participants with different work activity and different exposure to psychosocial risk factors could result in more conclusive findings. Notwithstanding, this study contributed to an area in need of more research and may be relevant for planning brief and effective interventions, preferably offered to nurses at their institutions in order to enhance health professionals’ health, namely preventing work-related musculoskeletal disorders.

## 6. Conclusions

Psychosocial risk factors exert a notable influence on work-related musculoskeletal disorders. Conducting research in this domain is crucial for understanding the psychosocial risk factors affecting nursing professionals and evaluating the less apparent connections between work and health. This study made significant contributions to better understand the impact of psychosocial risk factors on work-related musculoskeletal disorders during COVID-19.

Understanding the impact of age and regulatory processes developed by workers in specific work situations, in reality, reveals a balance between complying with work goals and preserving their health at work, namely in terms of work-related musculoskeletal disorders. Also, the integration of the worker’s perspective, expressed through perceptions, complaints, and feelings about their health, enables a more comprehensive and dynamic approach to occupational health. It is important to consider the discussion about improving health outcomes and underscores the significance of organizational support mechanisms aimed at enhancing working conditions, including restructuring work organization and fostering social support networks.

It is important, as demonstrated in this study, to concentrate on analysing the mechanisms responsible for these disorders. Additionally, establishing causality in the relationship between work and health is complex; therefore, further studies are warranted to enhance our understanding of this relationship and advocate for improved working practices, especially during alarming and other emergency situations. In this sense, the integration of an explicit reference to psychosocial risk factors in the promotion and prevention of occupational health and in the approach to public health involves making an explicit commitment to improving workers’ health.

## Figures and Tables

**Table 1 ijerph-21-00564-t001:** Descriptive analysis of the sample.

	Pre-Pandemic	Pandemic
Sample	N = 172	N = 284
**Gender**		
Female	84.8%	82.0%
Male	15.2%	18.0%
**Age (years)**		
M	37.52	38.85
SD	8.69	10.03
Range	24–57	21–64
**Experience years (n = 456)**		
M	11.22	13.49
SD	8.59	9.63
Range	1–35	1–42
**Work Contract**		
Full-time working hours	95.0%	93.3%
Permanent or open-ended contracts	91.0%	90.9%
Work rotating shifts	75.0%	65.1%

N—number of participants; M—mean; SD—standard deviation.

**Table 2 ijerph-21-00564-t002:** Distribution of “yes” responses concerning psychosocial risk factors in the workplace in pre-pandemic and pandemic times.

Psychosocial Risk Factors	Pre-Pandemic	Pandemic
Work pace and intensity	% Yes	% Yes
Working at an intense pace	98.1	94.4
Relying on direct orders from customers	90.5	76.1
Managing contradictory instructions	78.1	82.9
Frequent interruptions	86.7	82.4
Hyper-solicitation	88.6	81.0
Working Hours	% Yes	% Yes
Exceeding normal working hours	89.5	86.3
“Skipping” or shortening a meal or not taking a break at all because of work	94.3	82.7
Maintaining permanent availability	71.4	51.1
Work Relationships	% Yes	% Yes
Needing help from colleagues and not having it	48.3	46.1
My opinion being disregarded for the service’s functioning	54.4	44.4
Little recognition by management	49.2	47.5
Not having anyone I can trust	33.1	30.0
Being exposed to bullying	37.2	39.4
Employment relationships	% Yes	% Yes
Lack of means to perform the work	69.5	50.7
I feel exploited most of the time	66.7	50.0
Being afraid of suffering a work-related injury	82.9	71.1
The company disregards my well-being	74.3	63.0
Emotional demands	% Yes	% Yes
Confronted with tense public relations situations	94.3	94.7
Fear of verbal aggression	93.3	86.3
Being exposed to the difficulties and/or suffering of other people	98.1	94.7
Simulating good mood and/or empathy	83.8	75.4
Hiding emotions	88.6	75.0
Ethical and value conflicts	% Yes	% Yes
Doing things I disapprove of	71.4	64.8
My professional conscience is undermined	32.4	52.8
Lack of necessary means to perform a good job	65.7	57.7

**Table 3 ijerph-21-00564-t003:** Distribution of “yes” responses concerning musculoskeletal disorders in pre-pandemic and pandemic times.

	Pre-Pandemic		Pandemic	
	N	% Yes	N	% Yes
**Work-related musculoskeletal disorders**	172	56.2	284	63.2
Caused by my work activity	97	28.9	180	23.1
Aggravated by my work activity	97	33.4	180	56.9
Not related to my work activity	97	37.7	180	20.0

**Table 4 ijerph-21-00564-t004:** Logistic regression analysis to identify associations between psychosocial risk factors and musculoskeletal disorders and in pre-pandemic and pandemic times, adjusted for age.

Dimension	Psychosocial Risk Factors	*p*	OR_adjusted_ (C.I. 95%)
**Pre-Pandemic**			
Work pace and intensity	Managing contradictory instructions	<0.001	16.940 (3.289–87.249)
Working hours	Exceeding normal working hours	0.029	3.892 (1.745–18.324)
Emotional demands	Simulating good mood and/or empathy	0.009	5.480 (1.526–19.684)
**Pandemic**			
Employment relationships	Being afraid of suffering a work-related injury	<0.001	4.611 (2.090–10.172)
	My organization disregards my well-being	0.002	4.058 (1.703–9.673)
Ethical and values conflicts	Lack of essential means to perform a good job	0.016	2.502 (1.188–5.273)
Working relationships	Needing help from colleagues and not having it	0.003	2.994 (1.453–6.170)

*p—p* value; OR—odds ratio; C.I.—confidence interval.

**Table 5 ijerph-21-00564-t005:** Point biserial correlation coefficient for age and work-related musculoskeletal disorders.

WMSDs	*r*	*p*
**Pre-Pandemic**	−0.207	0.279
**Pandemic**	−0.334	0.025

WRMSDs—work-related musculoskeletal disorders; *r*—Pearson’s correlation coefficient; *p*—*p* value.

**Table 6 ijerph-21-00564-t006:** Point biserial correlation coefficient for age and psychosocial risk factors.

Dimension	Psychosocial Risk Factors	*r*	*p*
** *Pre-Pandemic* **			
Work pace and intensity	Managing contradictory instructions	−0.199	0.042
	Frequent interruptions	−0.268	0.006
	Hyper-solicitation	−0.312	0.001
**Pandemic**			
Work pace and intensity	Hyper-solicitation	−0.215	0.043
Employment relationships	Being afraid of suffering a work-related injury	−0.126	0.034
Emotional demands	Being exposed to the difficulties and/or suffering of other	−0.226	<0.001
	Simulating good mood and/or empathy	−0.206	0.046
	Hiding emotions	−0.195	0.049

*r*—Pearson’s correlation coefficient; *p*—*p* value.

## Data Availability

The data presented in this article are not readily available because they were not approved to be shared outside of the research team. Requests to access the datasets should be directed to cbarros@ufp.edu.pt or pilarbaylina@ess.ipp.pt.

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
