# Peer review of "Disclosing Strain: How Psychosocial Risk Factors Influence Work-Related Musculoskeletal Disorders in Healthcare Workers Preceding and during the COVID-19 Pandemicâ€"

_ijerph, 2024, doi:10.3390/ijerph21050564_

Round 1
Reviewer 1 Report
Comments and Suggestions for Authors
The article addresses a relevant phenomenon relating psychosocial factors and their relationships with MSDs in nurses, a healthcare professional category under stressful conditions. Another merit of the research was its trial to comparison of pre-pandemic and pandemic periods. However, there are some issues that the authors should clarify. Below, I list some of these points to guide authors in reviewing the text. 1. The title mentions, "How psychosocial factors influence work-related...". Notwithstanding, the survey design does not allow one to evaluate the process and infer the influence of one variable toward another. 2. The process and the influence between variables could be apprehended if the methodological design was based on repeated measures using the same sample. There were two independent samples. 3. The problem should be more justified regarding its theoretical and practical contributions. Particularly in the discussion section. 4. Regarding a theoretical reference, the authors did not consider the two psychosocial factors: protective and risky. Why the option only to the risky factors? 5. Why did the authors not mention COPSOQ I and II, which versions have validation in many countries, including Portugal? What is the advantage of the INSAT related to COPSOQ versions for measuring risky psychosocial factors? 6. In the same line, what is the advantage of using the Rasch Partial Credit Model instead of a Factorial analysis? 6. Tables 2 and 3 present a descriptive analysis of the INSAT survey. Were there any comparative statistics used to support the interpretations? 7. Based on the results, the authors should explore the study's practical implications for the nurses' health program. Finally, considering the many risk factors, how can human resources professionals prevent health damage to the nurses?
Reviewer 2 Report
Comments and Suggestions for Authors
Dear authors,
Thank you for this intersting study. However, I have some notes that you may consider in order to improve your paper.
Introduction- I suggest to improve the organization of the ideas in this section. Most of the statements are redundant and cyclical.
You need to mention from the start your thesis statement and from there you can discuss the points that support your arguments in your thesis statement. Example: Discuss about muscuoskeletal issues among nurses, then your argument (predictors) and the support to your arguments (why are these considered predictors).
There is also a need to consult an English editor who can help organize the sentence construction in case there is no one among the authors who can improve the current form.
What is the relevance of compating pre pandemic and pandemic situations when these are already obvious? These are two totally different situations and the difference is conspicous. You need to have a strong justification/ground for wanting to do this.
I also suggest for you to have a clear discussion of what is the gap you are trying to address in your study. What is your thesis statement, and what is your framework for psychological risks factors during pre-pandemic and pandemic times?
Materials and Methods:
Study design- What is the design of the study? Cross-sectional mainly informs about the point of time when data was collected but does not inform further how this design guided the achievement of the study aims.
Discussion about the data collection using INSAT should be under measures section or in a separate section for data collection and not under design. Thesame with the discussion about the population/participants of the study.
Measures:
What is the framework used as basis for the definition of psychological risks, psychological demands and efforts? I suggest that you include your citations/references for these concepts. And these should be clearly discussed in the introduction at the outset.
I suggest that you also improve discussion of the INSAT by starting with the meaning of INSAT, what does it measure, what are the dimensions included, how many items for each dimension. How is the instrument answered (likert scale? and describe the scores) ? Describe the scoring and interpretation of the scores. Include the validity and reliability of the tool.
The current format is not clear and is not easy for the reader to grasp.
Results:
3.1 Profile of the participants
-Age and years of work experience shpuld include the range and not only the mean and SD.
Informatin in lines 192 and 193 are not found in table 1 or elsewhere.
Table 2 The title should not be the analysis tool used but should be the variable being measured.
In the measures section- you indicated that the tools was answered using 6 point likert scale, how and why were these results reported as yes (%) for both pre pandemic and pandemic times?
How was the prevalence of work-related musculoskeletal disorders measured?
Your study focused on work-related musculoskeletal disorders and your table 3 should work-related, aggravated by work, and not realted with work- what measures were used in correlating the psychological factors to the work related musculoskeletal disorders?
Among all the items what were the items considered as factors? You just presented % of yes (of which from where this yes answers wetre taken? The scale used is 6-point likert scale). If the mean scores were used or median, the factors should have been identified.
Table 3 the title should not be the analysis tool used but should be the variables measured
The table 3 and the interpretation provided do not show a meaningful result.
Also what data were used in the logistic regression? You have participants with non-work related muscuoskeletal disorders.
Your aim to compare the risks and work-related musculoskeletal disordes between the pre pandemic and pandemoc was not statistically tested.
Conclusion:
What is new information about age as a factor to both psychological factors and musculoskeletal disorders?
What was the significant contribution of the study.
How do you contextualize your concusion for both pre-pandemic and pandemic times?
Thank you and good luck to you.
Comments on the Quality of English Language
There is a need to improve sentence constructions and organization of ideas to better communicate the meaning or focus of points being presented.
Reviewer 3 Report
Comments and Suggestions for Authors
Thanks to the authors for sharing their manuscript. Please accept my comments:
1. I would add ‘healthcare workers’ to the keywords.
2. The current title of the article is more suitable for describing a longitudinal study. I would think about reflecting the cross-sectional nature of the study in the title.
3. I would strengthen the introduction through a more thorough analysis of the changes in the physical and psychological functioning of healthcare workers that occurred during the COVID-19 pandemic.
4. The authors emphasize several times that the nurses participating in the study worked in private and public hospitals. Does the number of each subgroup allow for additional analysis depending on the work of private and public hospitals? Previous studies have shown that this factor may be important.
5. Although the research is cross-sectional, it seems to me possible to draw a conclusion about practical implications.
6. Technical comment: part of the text in the tables is written in blue, not black.
Reviewer 4 Report
Comments and Suggestions for Authors
The topic of the study is very relevant. Its findings are important not only for scientists, but also for employers, specialists of human resource management services and health care institutions. But the authors should revise and correct the manuscript before to print it.
The authors introduce the relevance of the topic in the Introduction, but they do not justify the novelty of their study. The relationship between psychosocial risk factors and the work-related musculoskeletal disorders among nurses are discussed in the previous studies, so why one more study is needed?
The purpose of the study and the chapters Results, Discussion and Conclusions should be revised and corrected. The study methods and design (cross-sectional research) does not let to evaluate one variable impact/influence to another.
More information how the selection of sample and institutions was carried out is needed in the manuscript.
The measurement instruments are not described clearly, there are no items examples of the questionnaires.
The results are really very interesting and provoking discussion but it seems the authors don't take advantage of it. For example, the big difference is on evaluation of the statement "My professional conscience is undermined" in pre-pandemic and pandemic period. It seems that this difference could be statistically significant but the authors did not do any statistical procedure to compare different results of pre-pandemic and pandemic period. Also there are no discussion about such kind of results in the Discussion part.
Round 2
Reviewer 1 Report
Comments and Suggestions for Authors
I greatly respect the authors' autonomy, but they could have been more receptive to incorporating changes in this case. They positioned themselves more in defense of the study they carried out than in respecting and trying to accept points for improvement suggested by the reviewer. Despite this disrespectful stance, the article can be published. In the future, I will not be available to review another manuscript from these authors.
Reviewer 4 Report
Comments and Suggestions for Authors
The study methods and design (cross-sectional research) does not let to evaluate one variable impact/influence to another. The text where is written about the impact/influence should be corrected. I wrote about that in my previous review, but the corrections were not made, see, for example, lines 107-110.
